# Suppressors of Cytokine Signaling Are Decreased in Major Depressive Disorder Patients

**DOI:** 10.3390/jpm12071040

**Published:** 2022-06-25

**Authors:** Nobuyuki Kobayashi, Shunichiro Shinagawa, Tomoyuki Nagata, Masahiro Shigeta, Kazuhiro Kondo

**Affiliations:** 1Department of Virology, The Jikei University School of Medicine, Tokyo 105-8461, Japan; kkondo@jikei.ac.jp; 2Department of Psychiatry, The Jikei University School of Medicine, Tokyo 105-8461, Japan; shinagawa@jikei.ac.jp (S.S.); t.nagata@jikei.ac.jp (T.N.); masa@jikei.ac.jp (M.S.)

**Keywords:** major depressive disorder, inflammation, cytokine, suppressor of cytokine signaling, SOCS1, SOCS2, SOCS3, interleukin 1β, interleukin 6, tumor necrosis factor alpha

## Abstract

There is strong evidence for an association between major depressive disorder (MDD) and inflammation. However, some studies have not observed an increase in inflammatory cytokines in MDD, and the mechanism behind this is unknown. In the present study, we evaluated MDD severity using the Montgomery–Åsberg Depression Rating Scale (MADRS) and quantified mRNA levels of the blood inflammatory cytokines interleukin (IL) 1β, IL-6 and tumor necrosis factor alpha (TNF-α), as well as negative regulators of cytokine signaling—comprising IL-10, IL-1RA, SOCS1, SOCS2 and SOCS3—in MDD patients (*n* = 36), with a focus on mild MDD, and normal controls (NC, *n* = 30). We also measured the serum levels of IL-1β and IL-6. Neither the blood mRNA nor the protein levels of inflammatory cytokines were significantly elevated in the MDD group compared with the NC group. However, we observed significant decreases in SOCS1, SOCS2 and SOCS3 mRNA in the MDD group compared to the NC group. A significant finding was a decrease in SOCS3 mRNA after remission from MDD, suggesting that SOCS3 is a trait marker in depressive symptoms. We consider that our findings would be useful in elucidating the pathophysiological mechanism of depression.

## 1. Introduction

Major depressive disorder (MDD), whose main symptoms are depressed mood and decreased interest, is a major cause of suicide [1,2]. The lifetime prevalence in the US is reported to be 16.2–20.6% [3,4]. MDD is a common disease and a major social problem, but its cause and mechanism of onset are unknown. This can be attributed to the fact that MDD has several pathological states and diverse clinical courses. Familial history, early-life abuse, stress factors, metabolic and autoimmune disorders are known risk factors of MDD [5]. In particular, MDD has a high prevalence in individuals with autoimmune disorders, and it is becoming clear that the dysregulation of homeostasis-maintaining actions in immune cells is associated with MDD onset [6].

Immunity has peripheral and central components. Innate immune myeloid cells (e.g., macrophages/monocytes and dendritic cells), lymphoid cells (e.g., natural killer (NK) cells) and lymphocytes (T and B cells) are mainly responsible for the former, and microglia for the latter [5]. Recent research has shown that an abnormal immune response occurs in MDD, which has been observed as changes in blood inflammation markers [7,8] and the activation of microglia in the brain [9].

A meta-analysis noted that in blood inflammatory marker protein measurements, levels of the representative cytokines interleukin (IL)-6 and tumor necrosis factor alpha (TNF-α) were elevated in MDD compared with normal controls (NC), while there was no significant change in IL-1β [8]. However, in terms of peripheral blood mononuclear cell (PBMC) gene expression, increases in IL-1β, IL-6 and TNF-α genes in MDD relative to NC were reported [10]. The reason for the discrepancy in the findings for IL-1β protein and gene expression in MDD is unknown.

Additionally, some studies have not found significant changes in inflammatory cytokines [8]. This may have been due to the influence of such factors as treatment, obesity, and blood pressure on blood inflammatory cytokines [5,11]. The timing of blood collection would also influence cytokine levels, with IL-6 and TNF-α increasing in the acute phase of disease and then decreasing after treatment [12]. Although inflammation does not increase in all MDD patients, inflammatory cytokines are considered to be a cause of depressive symptoms, and anti-inflammatory drugs are being studied as a treatment for MDD [13].

In view of the above, although it is widely believed that inflammatory cytokines are involved in the pathophysiology of MDD, the mechanism remains unclear. According to a meta-analysis, a relative lack of counter-regulatory immune mechanisms was considered to be a cause [8], but there is no direct evidence for this. In addition, the reason for discrepancies among individual studies and inconsistency in blood IL-1β protein levels and mRNA levels is not known. Furthermore, hardly any studies have measured both blood protein and mRNA levels for inflammatory cytokines, and it is not clear which cells produce the inflammatory cytokines whose levels change in MDD. Therefore, the purpose of the present study was to measure both protein and mRNA levels for the representative inflammatory cytokines IL-1β and IL-6 in Japanese MDD patients in order to clarify the influence of inflammatory cytokines in blood in MDD.

IL-10 [14] and IL-1RA [15], an agonist of IL-1β, and suppressors of cytokine signaling (SOCSs) are important negative regulators and are known to be factors in suppressing inflammatory cytokine production. Thus, the involvement of these factors in the elevation of inflammatory cytokines is important. However, while associations of IL-10 and IL-1RA with MDD have been reported, no study has examined the associations between SOCS and MDD. Among the eight known members of the SOCS family, in the present study, we focused on SOCS1, SOCS2 and SOCS3, which are frequently the subject of research [16].

Previously, many studies have targeted moderate or more severe MDD with a mean score of 20–30 on the Montgomery–Åsberg Depression Rating Scale (MADRS) [17,18], which indicates the severity of depressive symptoms [19,20]. However, nothing is known about an association of inflammatory cytokines with mild MDD. Since patients with mild to moderate depressive symptoms are the most numerous, a wider analysis including mild cases is required to further clarify immune abnormalities in MDD patients. Therefore, we focused on mild depression in the present study.

## 2. Materials and Methods

### 2.1. Participants and Clinical Assessment

The study enrolled 36 patients with MDD at the Jikei University Hospital (Tokyo) and the Jikei University Kashiwa Hospital (Chiba). Potential subjects were the same as those who participated in our previous research [21]. Among them, we selected those from whom serum and mRNA samples were collected as the present subjects. They were diagnosed using DSM-IV criteria by psychiatrists. A total of 30 volunteers from among staff of the Airanomori Hospital with no history of psychiatric consultation served as the NC group. Potential subjects presenting inflammatory-related comorbidities with fever and infection symptoms were excluded. Blood samples were collected from 9:00 to 11:30 a.m. Concurrently, depressive symptoms were evaluated in all participants using the Japanese version of the Beck Depression Inventory (BDI, range 0–63), a subjective measure of depressive symptoms [22]. Additionally, objective depressive symptoms in MDD were evaluated using the Japanese version of the Montgomery–Åsberg Depression Rating Scale (MADRS, range 0–60) by psychiatrists [19]. There were 15 MDD patients who had an MADRS score of 6 or less and were in remission (MDD-R) due to treatment [23] but were still under treatment. There were 21 MDD patients with an MADRS score of 7 or above (current MDD: MDD-C). Depression was still evident in these patients, and they remained under treatment. Individuals aged 65 years and over were excluded, and so were NC patients with BDI scores of 17 and above (recommended cutoff in screening with BDI [24]). As they were on humans, studies were approved by the Ethics Committees of the Jikei University School of Medicine (24-250 (7016) and 25-167 (7302)). Written informed consent was obtained from each subject.

### 2.2. Real-Time RT-PCR

Blood samples were collected in PAXgene Blood RNA tubes, and total RNA was extracted from whole blood using the PAXgene Blood RNA Kit (QIAGEN, Hilden, Germany). cDNA was synthesized from total RNA using the PrimeScript RT reagent Kit (Takara Bio, Otsu, Japan). Blood mRNA levels of IL-1β (Hs01555410_m1), IL-6 (Hs00985639_m1), TNF-α (Hs00174128_m1), IL-10 (Hs00961622_m1), IL-1RA (Hs00893626_m1), SOCS1 (Hs00705164_s1), SOCS2 (Hs00919620_m1), SOCS3 (Hs02330328_s1) and actin beta (ACTB) (Hs01060665_g1) were determined using TaqMan Array Cards with the Applied Biosystems QuantStudio 12 K Flex Real-Time PCR System (Thermo Fisher Scientific, Waltham, MA, USA). Following the instruction manual, amplifications were performed in duplicate, in a total volume of 100 µL per port containing 50 µL TaqMan Gene Expression Master Mix (Thermo Fisher Scientific, Waltham, MA, USA), 40 µL PCR-grade water and 10 µL cDNA. The reaction started with an initial denaturation step at 95 °C for 10 min, followed by 40 cycles of 15 s at 95 °C and 1 min at 60 °C. Data were analyzed using ExpressionSuite Software v1.1 (Thermo Fisher Scientific, Waltham, MA, USA).

### 2.3. Enzyme-Linked Immunosorbent Assay (ELISA)

Blood samples were centrifuged, and sera were stored frozen at −80 °C until assay. We used the Human IL-1 beta/IL-1F2 Quantikine HS ELISA Kit (HSLB00D, R & D Systems, Minneapolis, MN, USA) and Human IL-6 Quantikine ELISA Kit HS (HS600C, R & D Systems, Minneapolis, MN, USA) for measurements of IL-1β and IL-6 in sera. The sensitivity and range of these assay kits were 0.063 pg/mL and 0.1–8 pg/mL, and 0.09 pg/mL and 0.16–10 pg/mL, respectively. IL-1β and IL-6 in serum were measured in duplicate according to the attached protocols. A TriStar LB941-vTi Microplate Reader (Berthold Technologies, Bad Wildbad, Germany) was used to measure optical density (450 nm, reference 560 nm). The standard curves were obtained through cubic regression using the included highly purified E. coli-expressed recombinant human IL-1β and IL-6.

### 2.4. Statistical Analysis

The Shapiro–Wilk W test was used to test the normality of data. Parametric 2-group comparisons were conducted using the unpaired *t*-test. Non-parametric 2-group comparisons were conducted using the Mann–Whitney *U* test. The chi-square test was used for comparison by sex. The 3-group comparisons of non-parametric data were conducted using the Kruskal–Wallis test. In addition, if there were significant differences, Dunn’s multiple comparisons test was used for subsequent testing, as a post hoc test, as needed. Relationships between variables were assessed using Spearman’s rank correlation coefficients. *p* < 0.05 was considered statistically significant. Statistical analyses were conducted using SPSS Statistics 21 for Windows (IBM Corporation, Armonk, NY, USA) and Prism 9 for macOS (GraphPad Software, San Diego, CA, USA).

## 3. Results

### 3.1. Characteristics

Ages of MDD and NC subjects were assumed to be normally distributed, but body mass index (BMI), duration of disease, BDI score and MADRS score were not normally distributed. In Table 1, the mean is used for age and the median is used for other parameters except sex. Comparing the MDD group with the NC group, no significant differences were observed for age, sex or BMI. Compared to the NC group, the BDI score was significantly higher in the MDD group.

### 3.2. Associations of IL-1β, IL-6 and TNF-α mRNA Expressions with Depressive Symptoms

The mRNA levels of the inflammatory cytokines IL-1β, IL-6 and TNF-α were not normally distributed. In addition, overall, no significant differences in IL-1β, IL-6 and TNF-α mRNA were observed between the MDD group and the NC group (Figure 1A,D,G; *p =* 0.806, *p =* 0.962 and *p =* 0.776, respectively). With regard to the presence or absence of depressive symptoms, no significant differences were observed among the MDD-R, MDD-C and NC groups (Figure 1B,E,H; *p =* 0.748, *p =* 0.867 and *p =* 0.164, respectively), and there was no significant correlation between IL-1β or IL-6 mRNA and the MADRS score (Figure 1C,F; *p =* 0.757 and *p =* 0.119, respectively). The only significant correlation was between TNF-α mRNA and the MADRS score (Figure 1I).

### 3.3. Serum Concentrations of IL-1β and IL-6

The serum concentrations of IL-1β and IL-6 were not normally distributed. Compared to the NC group, the serum concentration of IL-1β was significantly lower in the MDD group (Figure 2A,B; *p* < 0.0001 and *p* < 0.0001, respectively). Moreover, the serum IL-1β concentration was positively correlated with the MADRS score (Figure 2C). However, due to the large number of negative values for concentration, which were below the detection limit, the results were lacking in reliability. Additionally, serum IL-1β and blood IL-1β mRNA were not correlated (Figure 2D; *p =* 0.702). In addition, serum concentrations of IL-6 in the MDD group were not significantly different compared to the NC group (Figure 2E,F; *p =* 0.975 and *p*
*=* 0.957, respectively). The serum IL-6 concentration and the MADRS score were not correlated (Figure 2G; *p =* 0.486). Serum IL-6 and blood IL-6 mRNA were also not correlated (Figure 2F; *p =* 0.665).

### 3.4. Associations of Negative Regulator of Cytokine Signaling mRNA Expression in Blood and Depressive Symptoms

IL-10, IL-1RA, SOCS1, SOCS2 and SOCS3 mRNA were not normally distributed. There was no significant difference in IL-10 or IL-1RA between the MDD group and the NC group (Figure 3A,D; *p =* 0.239 and *p =* 0.234, respectively). Compared to the NC group, SOCS1, SOCS2 and SOCS3 mRNA were significantly lower in the MDD group (Figure 3G,J,M; *p =* 0.045, *p =* 0.001 and *p =* 0.037, respectively). In addition, with regard to the presence or absence of depressive symptoms, a comparison among the MDD-R, MDD-C and NC groups revealed no significant differences in IL-10, IL-1RA or SOCS1 mRNA (Figure 3B,E,H; *p =* 0.496, *p =* 0.189 and *p =* 0.079, respectively), while SOCS2 mRNA was significantly lower in the MDD-C group than in the NC group (Figure 3K; *p =* 0.004). Furthermore, SOCS3 mRNA was significantly lower in the MDD-R group than in the NC group (Figure 3N; *p =* 0.026). However, there were no correlations of negative regulators with disease severity (Figure 3C,F,I,L,O; *p =* 0.913, *p =* 0.098, *p =* 0.326, *p =* 0.740 and *p =* 0.336, respectively).

### 3.5. Examination of Confounding Factors

Correlations of blood cytokines and negative regulators of cytokine signaling with age, BMI and disease duration are shown in Table 2. IL-1β mRNA was negatively correlated with duration of disease. There was a positive correlation between serum IL-6 and BMI, and IL-10 mRNA was negatively correlated with duration of disease. SOCS mRNA was not correlated with age, BMI or duration of disease.

## 4. Discussion

There is thought to be a major correlation between inflammation and MDD. In the present study, we examined the involvement of peripheral inflammation focusing on mild MDD patients. Although we did not observe an increase in inflammatory cytokine levels in these patients, mRNA levels of SOCSs, which are negative regulators of inflammation, decreased.

On comparing blood IL-1β, IL-6 and TNF-α mRNA levels in the MDD and NC groups, differing from previous research [10], we found no significant difference (Figure 1A–H). However, there was a significant correlation between TNF-α mRNA and MADRS score (Figure 1I), suggesting that blood inflammatory cytokine gene expression would be elevated in severe MDD.

In addition, no significant elevations in serum IL-1β and IL-6 were observed in the MDD group, compared with the NC group (Figure 2A,B,E,F). Although a meta-analysis reported that compared to NC, blood IL-6, but not IL-1β was elevated in MDD [8], we observed no increase in IL-6 mRNA or serum IL-6. In the present study, according to ELISA, many IL-1β values were negative (Figure 2A). The sensitivity and range of the IL-1β measurement ELISA kit we used were 0.063 pg/mL and 0.1–8 pg/mL, respectively, so if there had been no particular infection, the serum IL-1β concentration would have been lower than this, suggesting difficulty in quantification using current ELISA kits. A previous study reported no significant difference between MDD and NC for blood IL-1β concentration when protein was measured [8], while another found a significant difference in IL-1β mRNA levels between MDD and NC [10]. The first of these results may reflect our finding that the blood IL-1β concentration was too low to measure, and all of these findings suggest that compared to measuring protein for IL-1β, sensitivity is higher for mRNA.

A MADRS score of 0–6 is considered to indicate recovery; 7–19, mild depression; 20–34, moderate depression; and 35–60, severe depression [23]. The median MADRS score in the present study was 11.5 in the MDD group (Table 1), showing that our study was focused on mild MDD. It has been reported that exercise causes blood inflammatory cytokine levels to rise [25], that regular exercise suppresses TNF-alpha [26] and that obesity and ageing are associated with the elevation of blood inflammatory cytokine levels [27]. Thus, there are factors other than depression that alter cytokine levels, and we considered this to be a reason for the lack of a significant elevation in inflammatory cytokine levels in the MDD group relative to the NC group.

In addition, we found no correlations between serum concentrations and blood mRNA expressions for IL-1β or IL-6 (Figure 2D,H). mRNA in the blood is derived from mononuclear cells, such as monocytes and lymphocytes, while inflammatory cytokines are reportedly produced by T cells and macrophages in the blood, as well as microglia [28] in the brain, skeletal muscle [29] and adipocytes [30]. Therefore, we considered the serum cytokines to be a mixture of those derived from different sources. The correlation between serum IL-6 and BMI (Table 2) observed in the present study suggested production by adipocytes. mRNA in the blood is derived from immune cells within it. With the correlation of PBMC TNF-α mRNA with depressive symptom severity (Figure 1I) that we observed and a previously reported significant increase in PBMC inflammatory cytokine mRNA in MDD compared with NC [10], it has now been demonstrated that peripheral immunity is directly or indirectly involved in the pathophysiology of MDD.

Next, we focused on IL-10, IL-1RA, SOCS1, SOCS2 and SOCS3 as negative regulators. We discovered that blood SOCS1, SOCS2 and SOCS3 mRNA were significantly lower in the MDD group overall compared to the NC group (Figure 3G,J,M). Of particular interest, SOCS3 mRNA was significantly lower in the MDD-R group than in the NC group, while there was no significant difference between the MDD-C group and the NC group in this respect, suggesting that SOCS3 levels would rise in response to increased depression activity. This was considered to be a reason why association with SOCS mRNA was not reported in previous studies.

SOCS3 is a well-known feedback inhibitor of the Janus kinases-signal transducer and activator of transcription 3 signaling pathway [31], implying that during MDD remission, there would be a tendency for inflammatory cytokines to be produced due to a decrease in SOCS3 mRNA. It has been reported that in older adults who had no subjective depressive symptoms, there was no change in plasma IL-6 concentrations after influenza vaccination, but in older adults with depressive symptoms, plasma IL-6 concentrations increased after vaccination [32]. This is considered to signify that with a stimulus, changes in inflammatory cytokines readily occur; thus, supporting our findings. The results of the present study are considered be related to the likelihood of MDD recurrence as well as trait markers of MDD and should be useful in clarifying the pathophysiology of MDD.

It is not clear when decreases in SOCS3 mRNA occur, and there have previously been no reports of SOCS3 gene abnormalities in MDD. Additionally, we found no correlation between age or duration of MDD and SOCS3 mRNA levels (Table 2); therefore, it is not known whether SOCS3 mRNA would normalize afterwards. Thus, further studies that start before onset and observe the long-term course of depressive symptoms in MDD patients will be needed.

The first limitation of the present study is the small number of subjects. Moreover, the MDD group was heterogenous in that there were subjects with varying degrees of severity, including those in remission. Therefore, the results are preliminary and further research with an increased number of subjects will be needed in the future. Although there were statistically significant differences, multiple factors (SOCS1, SOCS2 and SOCS3) were measured, and similar results were obtained for them. Thus, although we could expect an increase in subjects to reduce the probability of β errors, we consider that the conclusions of our study would be the same. Second, the effect of treatment was not considered. However, since the issue of whether antidepressant drugs, such as selective serotonin reuptake inhibitors (SSRIs) and serotonin and noradrenaline reuptake inhibitors (SNRIs), have an anti-inflammatory effect remains unresolved [33]; at a minimum, we consider it unlikely that they would have a major impact on the results of the present study.

Our results suggest that a drop in SOCSs in MDD could be associated with a tendency for inflammatory cytokines to increase, and we believe that this is an important new finding concerning the pathophysiology of MDD onset and recurrence. Additionally, while we were unable to observe an increase in inflammatory cytokines in mild MDD, decreases in SOCSs may be observable, so they could be used as biomarkers in the future. However, since decreases in SOCSs were not associated with severity, this suggests usefulness as a diagnostic marker, rather than as a monitoring marker.

## 5. Conclusions

In the present study, obvious increases in inflammatory cytokines in terms of protein and mRNA levels were not observed in mild MDD patients, compared with NC. However, in the MDD group, there was a decrease in mRNA for SOCSs, which regulate inflammatory cytokines. As a particular finding, a decrease in SOCS3 mRNA in patients in remission was observed, suggesting that SOCS3 could be a trait marker indicating a tendency for inflammatory cytokine levels to rise. We believe that this observation may be related to the mechanism of onset of MDD. Additionally, these biomarkers would be useful for diagnosis of MDD and potential MDD and, therefore, have implications for personalized medicine and the prevention of inflammation–related MDD.

## Figures and Tables

**Figure 1 jpm-12-01040-f001:**
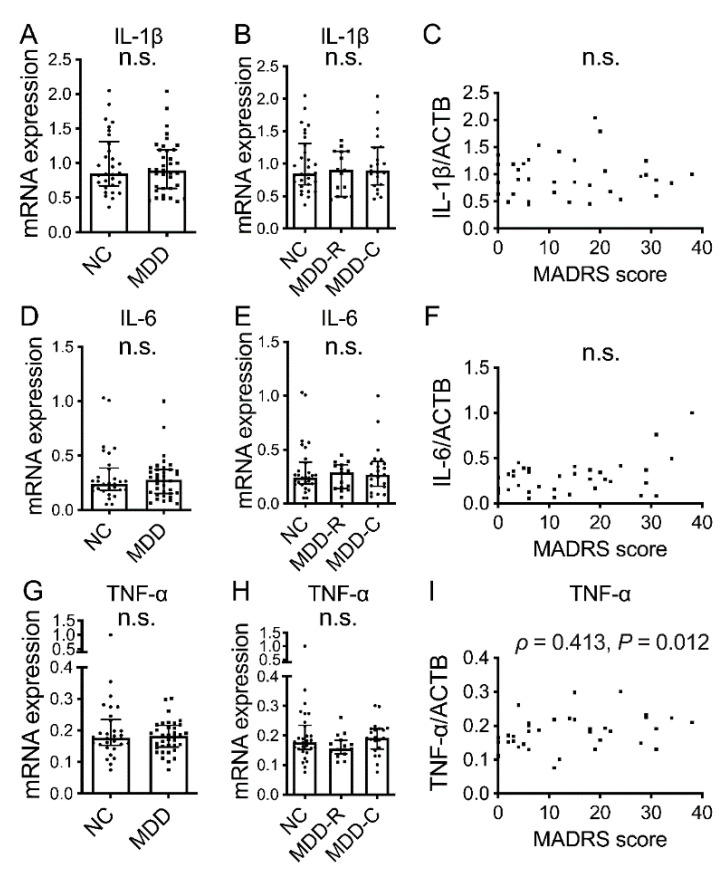
Blood mRNA for inflammatory cytokines in MDD and NC. Blood mRNA levels for IL-1β (**A**), IL-6 (**D**) and TNF-α (**G**) in NC group and MDD group overall. Blood mRNA levels for IL-1β (**B**), IL-6 (**E**) and TNF-α (**H**) in NC group, MDD remission group (MDD-R) and current MDD group (MDD-C). Horizontal lines indicate medians and error bars (IQR). Correlations of MADRS score with blood IL-1β mRNA (**C**), IL-6 mRNA (**F**) and TNF-α mRNA (**I**). n.s.: not significant.

**Figure 2 jpm-12-01040-f002:**
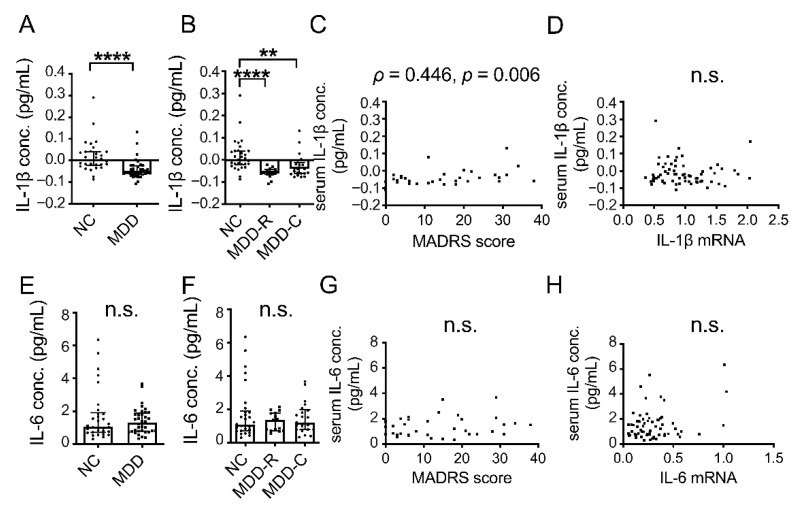
Serum concentrations of inflammatory cytokines in MDD and NC. Serum concentrations of IL-1β (**A**) and IL-6 (**E**) in NC group and MDD group overall. Serum concentrations of IL-1β (**B**) and IL-6 (**F**) in NC group, MDD remission group (MDD-R) and current MDD group (MDD-C). Horizontal lines indicate medians and error bars (IQR). ** *p* < 0.01. **** *p* < 0.0001. Correlation of MADRS score with serum concentration of IL-1β (**C**) and IL-6 (**G**). Correlation between blood mRNA and serum concentration for IL-1β (**D**) and IL-6 (**H**). n.s.: not significant.

**Figure 3 jpm-12-01040-f003:**
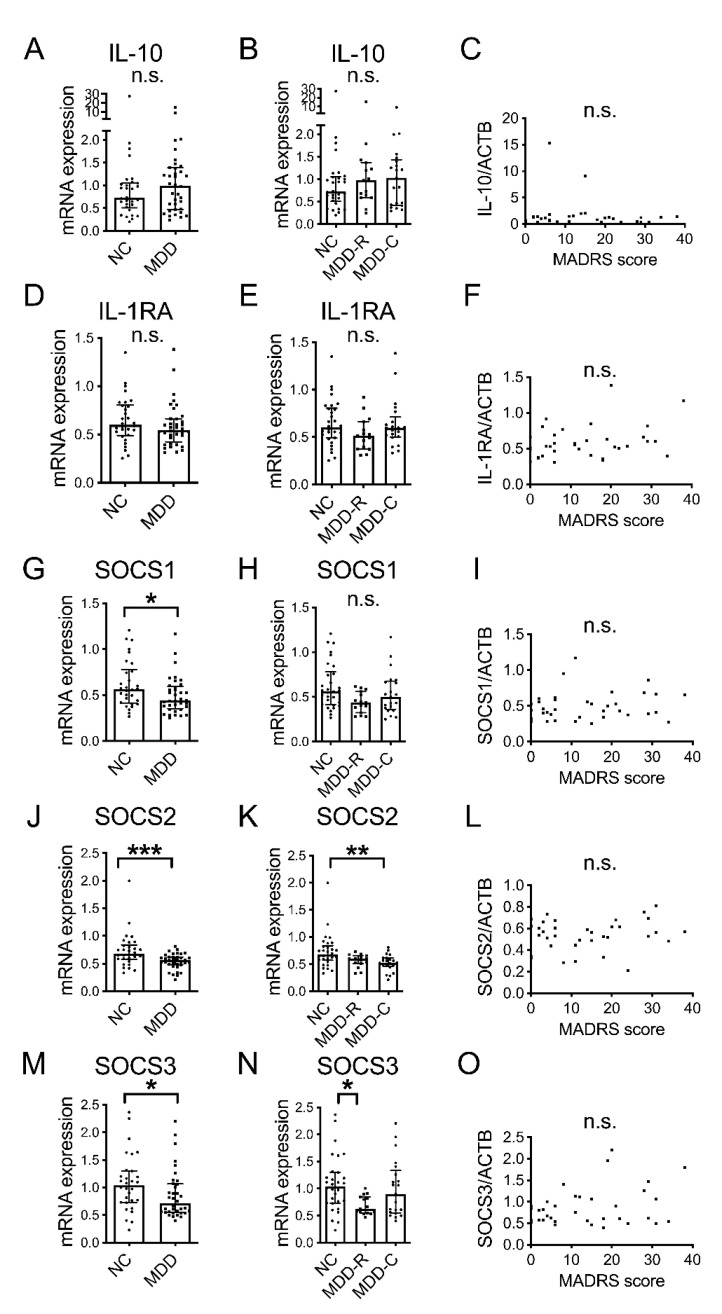
Blood mRNA for negative regulators of cytokine signaling in MDD and NC. Blood IL-10 mRNA (**A**), IL-1RA mRNA (**D**), SOCS1 mRNA (**G**), SOCS2 mRNA (**J**) and SOCS3 mRNA (**M**) in NC group and MDD group overall. Blood IL-10 mRNA (**B**), IL-1RA mRNA (**E**), SOCS1 mRNA (**H**), SOCS2 mRNA (**K**) and SOCS3 mRNA (**N**) in NC group and MDD-R group and MDD-C group. Horizontal lines indicate medians and error bars (IQR). * *p* < 0.05. ** *p* < 0.01. *** *p* < 0.001. Correlations of MADRS score with blood IL-10 mRNA (**C**), IL-1RA mRNA (**F**), SOCS1 mRNA (**I**), SOCS2 mRNA (**L**) and SOCS3 mRNA (**O**). n.s.: not significant.

**Table 1 jpm-12-01040-t001:** Characteristics of participants.

	NC (*n* = 30)	MDD (*n* = 36)	*p*
Age (years) (mean ± SEM)	44.9 ± 2.1	46.2 ± 1.8	0.643
Female:Male (%)	53.3:46.7	30.6:69.4	0.061
BMI (median (IQR))	23.2 (21.2−25.3)	23.6 (22.1−26.6)	0.438
Duration of disease (years) (median (IQR))	-	5.5 (2.0−10.8)	-
BDI scores (median (IQR))	2.0 (1.5−6.5)	12.5 (6.5−23.3)	0.000
MADRS scores (median (IQR))	-	11.5 (4.0−21.8)	-

SEM: standard error. IQR: interquartile range.

**Table 2 jpm-12-01040-t002:** Correlations of blood cytokines and negative regulators of cytokine signaling with age, BMI and duration of disease.

VS.		Age (*n* = 66)	BMI (*n* = 64)	Duration of Disease (*n* = 36)
IL-1β mRNA	*ρ*	−0.160	−0.085	−0.355
	*p*	0.200	0.505	0.034 *
IL-6 mRNA	*ρ*	0.054	0.131	−0.223
	*p*	0.668	0.302	0.191
TNF-α mRNA	*ρ*	0.144	0.017	−0.008
	*p*	0.247	0.897	0.962
serum IL-1β	*ρ*	0.019	0.019	−0.263
	*p*	0.881	0.880	0.121
serum IL-6	*ρ*	0.197	0.485	−0.143
	*p*	0.113	0.000 ****	0.405
IL-10 mRNA	*ρ*	−0.027	0.157	−0.344
	*p*	0.828	0.214	0.040 *
IL-1RA mRNA	*ρ*	0.009	−0.166	−0.137
	*p*	0.944	0.189	0.425
SOCS1	*ρ*	−0.060	−0.149	−0.087
	*p*	0.631	0.238	0.616
SOCS2	*ρ*	0.169	−0.014	−0.040
	*p*	0.174	0.911	0.819
SOCS3	*ρ*	−0.112	−0.187	−0.234
	*p*	0.371	0.138	0.170

* *p* < 0.05. **** *p* < 0.0001.

## Data Availability

Correspondence and requests for data should be addressed to N.K.

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
