# Peer review of "Suppressors of Cytokine Signaling Are Decreased in Major Depressive Disorder Patients"

_jpm, 2022, doi:10.3390/jpm12071040_

Round 1
Reviewer 1 Report
This paper explores the putative association between major depressive disorder (MDD) and inflammation.
The authors compare mRNA expression levels and serum concentrations of inflammation-related biomarkers on MDD patients with mild MDD (as evaluated by the Montgomery-Asberg Depression Rating Scale) and normal controls. While some known associations are not reproduced, the authors find significant increases in mRNA expression of Supressor of Cytokine Signaling genes in affected individuals. Serum concentrations of IL-1β were found to differ significantly between NC and MDD groups.
The paper is well structured and easy to follow, written in flawless English.
The statistical analysis is sound and the results are clearly discussed.
Small sample size is a limitation, which is identified and discussed by the authors.
One aspect of the paper is troubling: the serum concentrations of IL-1β, determined through ELISA come out negative.
This is not a major results and the authors clearly identify this issue and point out that this affects the reliability of the results.
This is not my area of expertise, so I would suggest that this issue is better addressed by any other referee with more experience in this subject.
The Conclusions section is somewhat terse.
While this is matter of style I would suggest expanding it, eventually linking these results to any potential implication in personalized medicine.
Author Response
Reviewer 1
This paper explores the putative association between major depressive disorder (MDD) and inflammation.
The authors compare mRNA expression levels and serum concentrations of inflammation-related biomarkers on MDD patients with mild MDD (as evaluated by the Montgomery-Asberg Depression Rating Scale) and normal controls. While some known associations are not reproduced, the authors find significant increases in mRNA expression of Supressor of Cytokine Signaling genes in affected individuals. Serum concentrations of IL-1β were found to differ significantly between NC and MDD groups.
The paper is well structured and easy to follow, written in flawless English.
The statistical analysis is sound and the results are clearly discussed.
Small sample size is a limitation, which is identified and discussed by the authors.
One aspect of the paper is troubling: the serum concentrations of IL-1β, determined through ELISA come out negative.
This is not a major results and the authors clearly identify this issue and point out that this affects the reliability of the results.
This is not my area of expertise, so I would suggest that this issue is better addressed by any other referee with more experience in this subject.
The Conclusions section is somewhat terse.
While this is matter of style I would suggest expanding it, eventually linking these results to any potential implication in personalized medicine.
R: We are very grateful for your careful reading of our manuscript and your useful comments. In response, we have made the revisions in Conclusions described in the following:
“Additionally, these biomarkers would be useful for diagnosis of MDD and potential MDD and, therefore, have implications for personalized medicine and prevention of inflammation–related MDD.” (lines 581-583)
Reviewer 2 Report
The authors compared expression of some proinflammatory cytokines (IL-1β, IL-6 and tumor necrosis factor alpha (TNF-α), and negative regulators and suppressors of cytokine signalling in the blood of major depressive disorder (MDD) patients and healthy volunteers. No differences were found in the cytokine mRNA and protein levels between the MDD and control groups. In contrast, the suppressors of cytokine signalling (SOCS1, SOCS2 and SOCS3) mRNAs were significantly reduced in the MDD patients. The obtained data, especially those regarding SOCS, are original and quite interesting. Methods are sound, but they should be more precisely described. At what times the blood samples were collected, a better description of inclusion and exclusion criteria, eg. inflammatory-related comorbidities is needed).
Specific remarks:
1. As mentioned by the authors, the major limitations of this study is the small number of patients and a possible effect of antidepressive treatment on the measured biochemical parameters. Moreover, the MDD group was rather heterogenous in terms of severity of depressive symptoms and occurence of remissions. Therefore, this study appears a bit preliminary.
2. A clinical significance of the main findings remains unclear, since no correlation of SOCS expression with disease severities were found.
3. At what times the blood samples were collected?
4. A more accurate description of inclusion and exclusion criteria, eg. inflammatory-related comorbidities, is needed).
5 The whole manuscript needs an extensive language revision and professional editing and should be sent to an expert in academic writing or professional service for proofreading. Many phrases should be corrected, e. g. “ In view of the above, although it is widely believed that inflammatory cytokines are involved in the pathophysiology of MDD, the mechanism remains unclear”.
Author Response
Reviewer 2
The authors compared expression of some proinflammatory cytokines (IL-1β, IL-6 and tumor necrosis factor alpha (TNF-α), and negative regulators and suppressors of cytokine signalling in the blood of major depressive disorder (MDD) patients and healthy volunteers. No differences were found in the cytokine mRNA and protein levels between the MDD and control groups. In contrast, the suppressors of cytokine signalling (SOCS1, SOCS2 and SOCS3) mRNAs were significantly reduced in the MDD patients. The obtained data, especially those regarding SOCS, are original and quite interesting. Methods are sound, but they should be more precisely described. At what times the blood samples were collected, a better description of inclusion and exclusion criteria, eg. inflammatory-related comorbidities is needed).
R: We are grateful that our manuscript has been favorably considered for publication as an Article and to Reviewer#1 for the detailed evaluation. In response, we have revised the manuscript accordingly and resubmit it now. We have addressed all of the comments in the revision.
Specific remarks:
- As mentioned by the authors, the major limitations of this study is the small number of patients and a possible effect of antidepressive treatment on the measured biochemical parameters. Moreover, the MDD group was rather heterogenous in terms of severity of depressive symptoms and occurence of remissions. Therefore, this study appears a bit preliminary.
R: We agree that our study is preliminary and have stated this as a limitation as follows:
“Moreover, the MDD group was heterogenous in that there were subjects with varying degrees of severity, including those in remission. Therefore, the results are preliminary and further research with an increased number of subjects will be needed in the future.” (lines 552-555)
- A clinical significance of the main findings remains unclear, since no correlation of SOCS expression with disease severities were found.
R: This comment regarding the absence of a correlation between a decrease in SOCS expression and symptom severity in MDD is very valid. Our findings suggest that SOCS expression could be useful as a diagnostic marker, rather than as a monitoring marker. Moreover, SOCSs could be involved in the mechanisms of MDD onset and recurrence. For clarification, we have modified the text below in Discussion.
“Our results suggest that a drop in SOCSs in MDD could be associated with a tendency for inflammatory cytokines to increase and we believe that this is an important new finding concerning the pathophysiology of MDD onset and recurrence. Additionally, while we were unable to observe an increase in inflammatory cytokines in mild MDD, decreases in SOCSs may be observable, so they could be used as biomarkers in the future. However, since decreases in SOCSs were not associated with severity, this suggests usefulness as a diagnostic marker, rather than as a monitoring marker.” (lines 565-571)
- At what times the blood samples were collected?
R: Blood samples were collected from 9:00-11:30 am. To clarify this, the text below in
- Materials and Methods, 2.1. Participants and Clinical Assessment has been modified as shown below.
“Blood samples were collected from 9:00 to 11:30 am. Concurrently, depressive symptoms were evaluated in all participants using the Japanese version of the Beck Depression Inventory (BDI, range 0-63), a subjective measure of depressive symptoms [22].” (lines 104-107)
- A more accurate description of inclusion and exclusion criteria, eg. Inflammatory-related comorbidities, is needed).
R: To provide further details of exclusion criteria, we have stated that those presenting inflammatory-related comorbidities with fever and infection symptoms were excluded in 2.1. Participants and Clinical Assessment, as below.
“Potential subjects presenting inflammatory-related comorbidities with fever and infection symptoms were excluded.” (lines 103-104)
- The whole manuscript needs an extensive language revision and professional editing and should be sent to an expert in academic writing or professional service for proofreading. Many phrases should be corrected, e. g. “In view of the above, although it is widely believed that inflammatory cytokines are involved in the pathophysiology of MDD, the mechanism remains unclear”.
R: In line with your comment, we have submitted our manuscript to MDPI for professional editing (English Editing ID: english-46236) and amended it accordingly.
Round 2
Reviewer 2 Report
The authors have addressed my remareks and the manuscript has been improved.